# Restored TDCA and valine levels imitate the effects of bariatric surgery

Markus Quante[1,2†], Jasper Iske[1,3†], Timm Heinbokel[1,4†], Bhavna N Desai[5], Hector Rodriguez Cetina Biefer[1,6], Yeqi Nian[1,7], Felix Krenzien[8], Tomohisa Matsunaga[1,9], Hirofumi Uehara[1,9], Ryoichi Maenosono[1,9], Haruhito Azuma[9], Johann Pratschke[8], Christine S Falk[3], Tammy Lo[10], Eric Sheu[10], Ali Tavakkoli[10], Reza Abdi[11], David Perkins[12], Maria-Luisa Alegre[13], Alexander S Banks[5], Hao Zhou[1], Abdallah Elkhal[1], Stefan G Tullius[1]*

[1]Division of Transplant Surgery & Transplant Surgery Research Laboratory, Brigham and Women's Hospital, Harvard Medical School, Boston, United States; [2]Department of General, Visceral and Transplant Surgery, University Hospital Tübingen, Tübingen, Germany; [3]Institute of Transplant Immunology, Hannover Medical School, Hannover, Lower Saxony, Germany; [4]Department of Pathology, Charité Universitätsmedizin Berlin, Berlin, Germany; [5]Division of Endocrinology, Diabetes and Metabolism, Beth Israel Deaconess Medical Center, Boston, United States; [6]Department of Cardiovascular Surgery, Charité Universitätsmedizin Berlin, Berlin, Germany; [7]Department of Urology, The Second Xiangya Hospital, Central South University, Changsha, China; [8]Department of Visceral, Abdominal and Transplantation Surgery, Charité Universitätsmedizin Berlin, Berlin, Germany; [9]Department of Urology, Faculty of Medicine, Osaka Medical and Pharmaceutical University, Osaka, Japan; [10]Division of Gastrointestinal and General Surgery, Department of Surgery, Brigham and Women's Hospital, Harvard Medical School, Boston, United States; [11]Renal Division, Brigham and Women's Hospital, Harvard Medical School, Boston, United States; [12]Division of Nephrology, Department of Medicine, University of Illinois at Chicago, Chicago, United States; [13]Department of Medicine, Section of Rheumatology, The University of Chicago, Chicago, United States

*For correspondence:
stullius@bwh.harvard.edu

†These authors contributed equally to this work

Competing interests: The authors declare that no competing interests exist.

## Abstract

**Background:** Obesity is widespread and linked to various co-morbidities. Bariatric surgery has been identified as the only effective treatment, promoting sustained weight loss and the remission of co-morbidities.

**Methods:** Metabolic profiling was performed on diet-induced obese (DIO) mice, lean mice, and DIO mice that underwent sleeve gastrectomies (SGx). In addition, mice were subjected to intraperitoneal (i.p.) injections with taurodeoxycholic acid (TDCA) and valine. Indirect calorimetry was performed to assess food intake and energy expenditure. Expression of appetite-regulating hormones was assessed through quantification of isolated RNA from dissected hypothalamus tissue. Subsequently, i.p. injections with a melanin-concentrating hormone (MCH) antagonist and intrathecal administration of MCH were performed and weight loss was monitored.

**Results:** Mass spectrometric metabolomic profiling revealed significantly reduced systemic levels of TDCA and L-valine in DIO mice. TDCA and L-valine levels were restored after SGx in both human and mice to levels comparable with lean controls. Systemic treatment with TDCA and valine induced a profound weight loss analogous to effects observed after SGx. Utilizing indirect

calorimetry, we confirmed reduced food intake as causal for TDCA/valine-mediated weight loss via a central inhibition of the MCH.

**Conclusions:** In summary, we identified restored TDCA/valine levels as an underlying mechanism of SGx-derived effects on weight loss. Of translational relevance, TDCA and L-valine are presented as novel agents promoting weight loss while reversing obesity-associated metabolic disorders.

**Funding:** This work has been supported in part by a grant from NIH (UO-1 A1 132898 to S.G.T., DP and MA). M.Q. was supported by the IFB Integrated Research and Treatment Centre Adiposity Diseases (Leipzig, Germany) and the German Research Foundation (QU 420/1-1). J.I. was supported by the Biomedical Education Program (BMEP) of the German Academic Exchange Service (DAAD). T.H. (HE 7457/1-1) and F.K. (KR 4362/1-1) were supported by the German Research Foundation (DFG). H.R.C.B. was supported the Swiss Society of Cardiac Surgery. Y.N. was supported by the Chinese Scholarship Council (201606370196) and Central South University. H. U., T.M. and R.M. were supported by the Osaka Medical Foundation. C.S.F. was supported by the German Research Foundation (DFG, SFB738, B3).

## Introduction

Obesity is a global epidemic with broad clinical and economic consequences. Data by the Global Burden of Disease Study estimate that overweight and obesity are causing 3.4 million deaths and 3.8% of disability-adjusted life years (DALYs) worldwide (*Lim et al., 2012*). Moreover, obesity is promoting numerous disorders including diabetes, cardiovascular disease, and cancer (*Saltiel and Kahn, 2001*; *Khandekar et al., 2011*). Therapeutic approaches including diets or approved pharmacological interventions treating obesity and related co-morbidities have only had limited success (*Yanovski and Yanovski, 2014*; *Gloy et al., 2013*).

Bariatric surgery including sleeve gastrectomies (SGx) has been successful in achieving a sustained body weight loss (*Chang et al., 2014*). Moreover, these procedures also display an effective treatment of obesity-associated type 2 diabetes (T2D) (*Schauer et al., 2014*) with durable HbA1C remissions (*Arterburn et al., 2013*), improved insulin sensitivity, and glycemic control (*Buchwald et al., 2004*), all resulting into long-term reduction of overall mortality and obesity- related risk factors (*Gloy et al., 2013*; *Sjöström, 2013*).

SGx has been conceptualized to elicit weight loss by physically restricting gastric capacity (*Gumbs et al., 2007*). Moreover, the procedure leads to a weight-independent reduction in plasma triglyceride levels (*Stefater et al., 2011*), improved HDL levels (*Benaiges et al., 2011*), increased hepatic insulin sensitivity (*Chambers et al., 2011*), and weight-independent alterations of gut-derived anti-diabetic hormones (GLP-1 and PYY; *Peterli et al., 2009*), implicating an altered metabolic profile.

At a molecular level, bile acid signaling is thought to constitute a mechanistic underpinning of bariatric surgery-induced weight loss. Bile acids have been shown to activate nuclear transcription factors involved in hepatic glucose metabolism (*Parks et al., 1999*) that promote GLP-1 secretion via signaling by the bile acid receptor TGR5 (*Thomas et al., 2009*). Moreover, bile acids have been identified to be essential for SGx-derived weight loss through signaling via the bile acid receptor FXR (*Ryan et al., 2014*). Consistently, various studies have reported increasing serum bile acids following SGx in experimental and clinical studies (*Myronovych et al., 2014*; *Albaugh et al., 2015*).

Thus, durable weight loss and amelioration of obesity-associated disorders after SGx are based on a complex metabolic and hormonal homeostatic circuitry rather than surgery-induced malabsorption alone. Clearly, understanding the mechanisms of bariatric surgery in detail and achieving weight loss and metabolic changes through a non-surgical treatment may represent a desired alternative.

Here, we made use of diet-induced obese (DIO) mice, a well-established model of obesity and performed SGx to delineate how bariatric surgery promotes long-term metabolic changes and the reduction of body weight.

We identified restored levels of taurodeoxycholic acid (TDCA) and L-valine following SGx in our experimental model and confirmed similar effects in patients undergoing SGx.

Administration of both, TDCA and valine in DIO mice, caused a robust and sustained weight loss, reduced fat tissue, and reversed DIO-associated T2D based on a decreased food intake mediated

through suppression of the hypothalamic orexigenic peptide melanin-concentrating hormone (MCH) intake.

# Materials and methods

**Key resources table**

| Reagent type (species) or resource | Designation | Source or reference | Identifiers | Additional information |
|---|---|---|---|---|
| Strain, strain background (*Mus musculus*) | Diet-induced obese (DIO) C57BL/6 mice | Taconic | B6-M | |
| Strain, strain background (*Mus musculus*) | Lean C57BL/6 mice | Taconic | DIO-B6-M | |
| Strain, strain background (*Rattus norvegicus*) | Wistar rats | Charles River | 003 | |
| Peptide, recombinant protein | Recombinant MCH | Cayman Chemical | ID: 24462 | |
| Commercial assay or kit | Direct-zol RNA MiniPrep kit | Zymo Research | ID: 205311 | |
| Commercial assay or kit | Reverse transcriptase QuantiTech RT Kit | Qiagen | ID: R2061 | |
| Commercial assay or kit | SYBR Green master mix | Applied Biosystems | ID: 4309155 | |
| Chemical compound, drug | MCHR1-I | Takeda | | |
| Chemical compound, drug | TDCA | Sigma-Aldrich | ID: T0875-25G | |
| Chemical compound, drug | Valine | Sigma-Aldrich | ID: V0513-25G | |
| Software, algorithm | GraphPad Prism software | San Diego, CA | SCR_002798 | |
| Software, algorithm | MetaboAnalyst 3.0 | Genome Canada | SCR 015539 | |
| Other | Glucose | Sigma-Aldrich | ID: 50-99-7 | |
| Other | HFD D12492 | Research Diets Inc | ID: 50-99-712492 | |

## Animals

Animal use and care were in accordance with institutional and National Institutes of Health guidelines. DIO C57BL/6 mice and lean littermates were purchased from Taconic (Taconic Farms Inc, Germantown, NY) for all studies. The study protocol was approved by the Brigham and Women's Hospital (BWH) Institutional Animal Care and Use Committee (IACUC) animal protocol (animal protocol 2016N000371). Obesity was induced by feeding animals ad libitum with a high-fat diet (HFD) that provides 60% of total energy as fat (D12492 diet, Research Diets Inc, New Brunswick, NJ) starting at 6 weeks of age for a duration of 12 weeks. Wistar rats used for intracerebral administration of recombinant MCH were group-housed and fed D12492 for 12 weeks (60% k/cal diet). Rats subsequently underwent cranial surgery placing a resealable canula in the lateral cerebral ventricle. All animals were maintained in specific pathogen-free conditions at the BWH animal facility in accordance with federal, state, and institutional guidelines. Animals were maintained on 12 hr light, 12 hr dark cycle in facilities with an ambient temperature of 19–22°C and 40–60% humidity and were allowed free access to water and standard chow. Euthanasia was performed by cervical dislocation following anesthesia with isoflurane (Patterson Veterinary, Devens, MA).

## Bariatric surgery

A gastric sleeve was created along the lesser curvature by transecting the stomach. The sleeve was then hand-sewn, using an 8–0 continuous Prolene suture. Sham animals underwent a laparotomy, the stomach was isolated, and blunt pressure was applied with forceps for corresponding durations.

## Metabolic experiments

- Conservative weight loss

DIO-C57BL/6 mice obese animals that were on an HFD (12492, Research Diets Inc) at 6 weeks of age for a duration of 12 weeks were then switched to a normal chow diet and consecutive body weight was assessed.

- Weight loss following bariatric surgery

DIO-C57BL/6 mice obese animals that were on an HFD (12492, Research Diets Inc) by 6 weeks of age for a duration of 12 weeks underwent sleeve gastrectomies or sham surgery. Subsequent to the surgery, animals had free access to water and a liquid diet for 2 days. On post-operative day 3, animals were switched back to a solid HFD again.

- Intraperitoneal (i.p.) glucose tolerance test

Glucose (2 g/kg, Sigma-Aldrich, St. Louis, MO) was injected for glucose tolerance testing which was conducted by 8 hr of daytime fasting. Blood glucose was monitored using an Accu-Check Aviva Plus blood glucose meter (Roche, Plainfield, IN).

- Metabolomic treatment

TDCA (50 mg/kg) and L-valine (200 mg/kg) (both from Sigma-Aldrich, St. Louis, MO) were dissolved in sterile ddH$_2$O and simultaneously administered by i.p. injection.

- MCH receptor one inhibitor (MCHR1-I) administration

MCHR1-I was obtained from Takeda, Chuo-ku, Tokyo, Japan. MCHR1-I was administered at 10 mg/kg via oral gavage. As vehicle solution 0.5% Methocel, 0.1% Tween 80, 99.4% distilled water were used. For combinatory treatment of MCHR1-I and TDCA/valine, mice were additionally injected intraperitoneally with TDCA/valine.

- Recombinant MCH (rMCH) administration

rMCH (Cayman Chemical) was dissolved in sterile ddH$_2$O (1 µg/µl) and 5 µl was administered by intracranial injection over 30 s into the lateral ventricle.

## Indirect calorimetry

Twelve DIO mice (control = 6, treatment = 6) were placed into the Columbus Instruments Comprehensive Lab Animal Monitoring System and maintained for 6 days. They were kept on HFD at 22°C ± 1°C ambient temperature for the duration of the experiment. After 1 day of acclimation, injections of TDCA/valine were performed at 2 p.m. each day for 5 days. Time graphs represent hourly averages throughout the experiment. Bar graphs correspond to the total, light, and dark cycles (12 hr cycles beginning and ending at 6 a.m. and 6 p.m.). Error bars represent SEM. Student's t-tests were performed on all bar graphs.

## Human samples

Serum samples from patients prior to and 3 months post sleeve gastrectomy were obtained with approval of the BWH Institutional Review Board and through cooperation with Dr Eric G. Sheu and the Center for Metabolic and Bariatric Surgery at BWH. Informed consent was obtained from all patients and samples were collected following BWH ethical regulations. Whole blood samples were obtained at routinely scheduled pre-operative and post-operative appointments, centrifuged to obtain serum, and then stored at −80°C up until metabolite measurements and data analysis exactly as described above for murine samples.

## Metabolite measurements by LC–MS/MS and data analysis

Whole blood samples were centrifuged at 13,000 $\times$ $g$ for 10 min at 4°C, and 200 µl of the supernatant were saved; 800 µl of cooled methanol (−80°C) were added to the supernatant for a final 80% (vol/vol) methanol solution. Samples were incubated for 6 hr at −80°C and then centrifuged at 13,000 $\times$ $g$ for 10 min at 4°C. Supernatants were collected, dried in a SpeedVac (Savant AS160, Farmingdale, NY), and stored at −80°C until analysis. Each sample was resuspended in 20 µl of LC/MS grade water and then analyzed with a 5500 QTRAP, a hybrid triple quadrupole/linear ion trap mass spectrometer, using a quantitative polar metabolomics profiling platform with selected reaction monitoring that covers all major metabolic pathways. An unbiased quantitative analysis of 260 detected metabolites was performed utilizing the web-based MetaboAnalyst 3.0 software examining high-throughput metabolomics data aiming to identify patterns that were significantly different between our experimental groups in three consecutive steps. First, heat map provided an initial assessment on the distribution of peak intensities in our three experimental groups. Next, significance analysis of microarray (SAM) revealed candidates with statistically significant differences among our three experimental groups (obese-lean-SGx). Finally, we utilized the statistic tool of the pattern hunter to search for the 'ideal compound candidate' with lowest intensities in obese animals and highest in animals after SGx. Absolute quantifications of TDCA and valine concentrations were generated using standard curves with known concentrations of TDCA and valine, respectively.

## Dissection of hypothalamus tissue, RNA extraction, real-time PCR

Mice were assigned to the following experimental groups, with n = 6 in each group: lean PBS-treated mice, lean TDCA/valine-treated mice, DIO-obese PBS-treated mice, DIO-obese PBS-treated mice fasted for 12 hr before tissue procurement, DIO-obese TDCA/valine-treated mice, DIO-obese TDCA/valine-treated mice fasted for 12 hr before tissue procurement, DIO-obese mice undergoing sleeve gastrectomy. Cage beddings were pooled and redistributed at days −6, –4, and −2 to normalize microbial flora among experimental groups. At day 0, treatment began and SGx were performed. Mice were treated for 13 days, with procurement of tissues performed on day 14. Animals were anesthetized with ketamine and sacrificed by decapitation. Brains were removed, and hypothalamus tissue was dissected and flash-frozen in liquid nitrogen. RNA was isolated using Direct-zol RNA MiniPrep kit (Zymo Research, Irvine, CA). cDNA was made from isolated RNA using oligo (dt), random hexamer primers, and reverse transcriptase QuantiTech RT Kit (Qiagen, Germantown, MD). Quantitative PCR was performed using the 7800HT (Applied Biosystems, Foster City, CA) thermal cycler and SYBR Green master mix (Applied Biosystems). Relative mRNA abundance was calculated and normalized to levels of the housekeeping gene cyclophilin A.

## Statistics

Unless otherwise specified in figure legends, comparisons between experimental groups were performed using Student's t-test. Survival curves were compared by using the log rank test. When applicable, mice were randomly assigned to treatment or control groups. All results were generated using GraphPad Prism software (San Diego, CA). A p-value of 0.05 was considered statistically significant.

# Results

## Bariatric surgery induces sustained weight loss by restoring the metabolite profile in both, mice and humans

Bariatric surgery is effective in 80–90% of obese individuals and leads to sustained weight loss and significant improvement of co-morbidities (*Gloy et al., 2013*; *Chang et al., 2014*; *Sjöström, 2013*). To assess mechanisms of surgically induced weight loss through SGx, we made use of C57BL/6 wild-type DIO mice, a well-established murine model of obesity (*Winzell and Ahrén, 2004*; *Wang and Liao, 2012*; *Figure 1A*).

Subsequent to SGx, we observed a significant weight loss (−34% by 2 weeks). In marked contrast, sham-operated DIO mice re-gained their pre-operative weight by post-operative day 14 (−5%), indicating that the observed weight loss was independent of the surgical trauma (*Figure 1B*). Thus, SGx promoted durable weight loss independent of the surgical procedure or dietary effects.

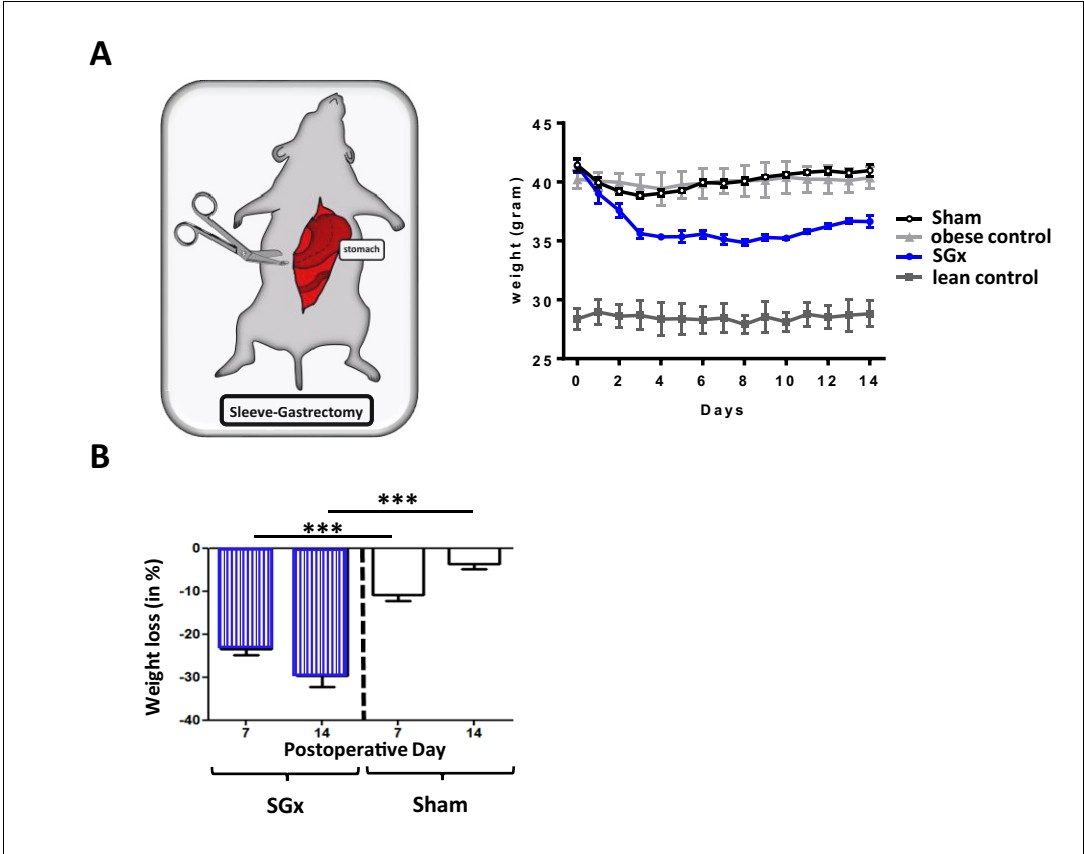

**Figure 1.** Sleeve gastrectomy (SGx) induces significant weight loss independent of the surgical procedure. C57BL/6 DIO mice (n = 5) underwent SGx or sham surgery, were fasted on the day of surgery, switched to a liquid diet for 2 days and returned to a high-fat diet by day 3. An additional set of C57BL/6 DIO and C57BL/6 lean mice receiving a similar diet served as controls. (**A**) Body weight was monitored for a course of 2 weeks every 24 hr. (**B**) Proportional weight loss of C57BL/6 DIO mice undergoing SGx and sham surgery was calculated comparing mean weight loss after 7 and 14 days, respectively. Results are representative of at least three independent experiments. Column plots display mean with standard deviation. Statistical significance was determined using two-way analysis of variance (ANOVA) followed by Turkey's multiple comparison test with single pooled variance. Asterisks indicate p-values: *p < 0.05, **p < 0.01, and ***p < 0.001. Only significant values are shown (n = 7 animals/group). The online version of this article includes the following source data for figure 1:

**Source data 1.** The numerical data for the graphs in *Figure 1*.

Previous studies implicated that SGx induce modifications in levels of metabolic and intestinal hormones (*Stefater et al., 2012*). Therefore, we performed a quantitative metabolomic profiling to characterize the consequences of obesity and weight loss surgery on endogenous metabolites. Our unbiased approach detected 260 metabolites with 17 candidates showing a significant statistical difference between lean and DIO mice by analysis of variance (ANOVA), heat mapping, and SAM (*Figure 2A,B*).

We found significantly decreased systemic levels of TDCA and L-valine in DIO mice. Obese animals, that underwent SGx, however, displayed overall restored levels of both metabolites, suggesting a critical involvement of TDCA/valine in the metabolic underpinning of SGx-induced weight loss (*Figure 2C*).

Moreover, to investigate the translational relevance of our findings, we analyzed serum levels of TDCA and valine in human samples and collected from patients immediately prior to and 3 months after SGx (*Figure 2D*). Subsequent to clinical SGx, we observed a significant increase in both TDCA and valine levels, indicating a similar impact of SGx in mice and humans (*Figure 2D*).

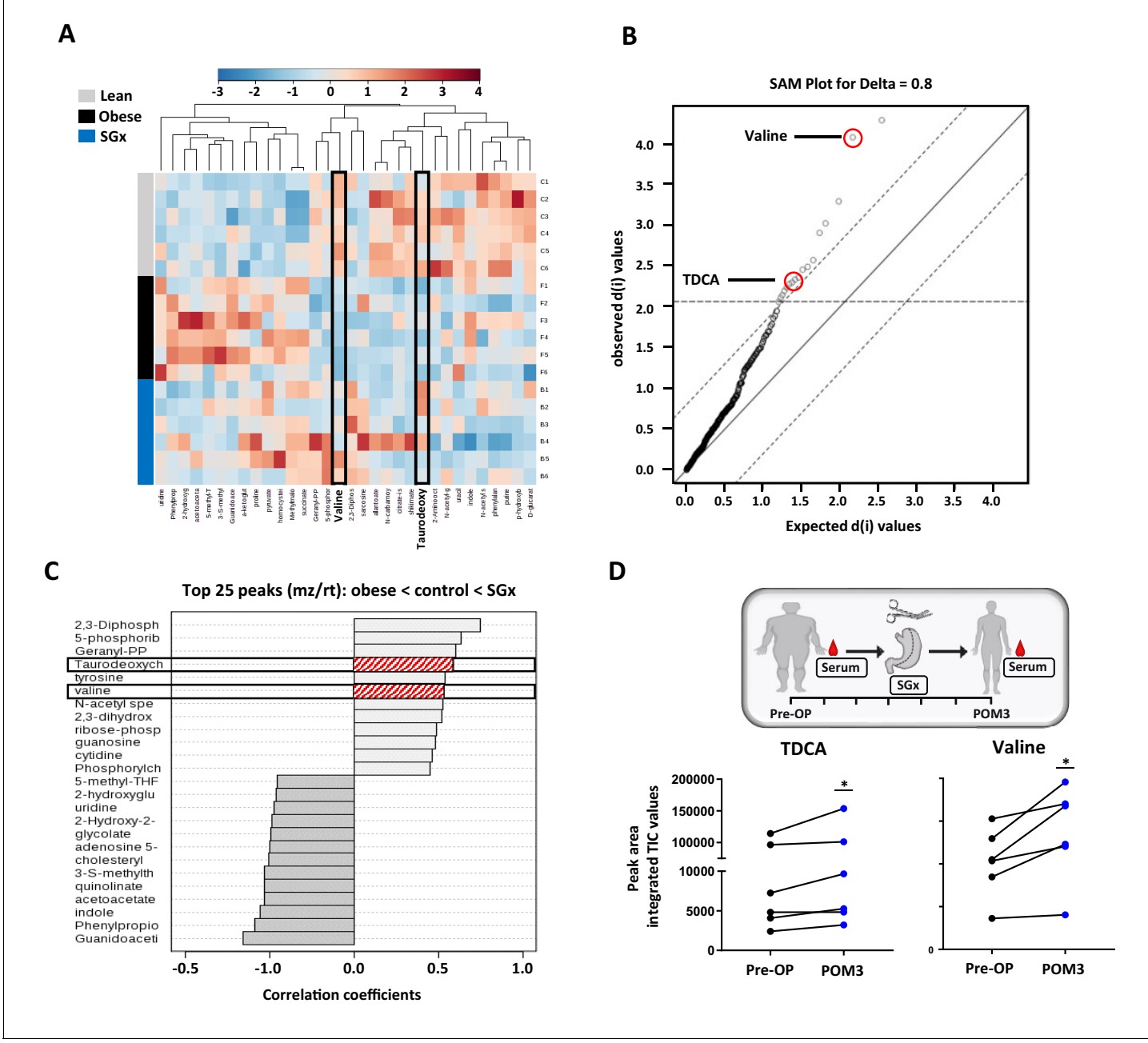

**Figure 2.** Sleeve gastrectomy (SGx) restores systemic taurodeoxycholic acid (TDCA)/valine levels in both diet-induced obese (DIO) mice and obese humans. Whole blood samples from C57BL/6 DIO mice after SGx and DIO and lean controls were analyzed with a 5500 QTRAP mass spectrometer. Quantitative analysis was performed utilizing MetaboAnalyst 3.0. (A) Heat map of 32 metabolites displayed after hierarchical clustering, p<0.05. (B) Significance analysis of microarrays (SAM) revealed 17 metabolites with significance. (C) Pattern hunter stratified the 25 metabolites with top peaks (mz/rt) according to the order obese-control-SGx. (D) Serum was isolated from patients undergoing SGx pre-operative (Pre-OP) and 3 months after surgery (POM3). TDCA and valine levels were quantified using mass spectrometry and peak are integrated TIC values compared (n = 6). Results are representative of at least three independent experiments. Statistical significance was determined using one-way analysis of variance (ANOVA) and SAM. TDCA/valine TIC values from human samples were compared using paired Student's t-test. Asterisks indicate p-values: *p < 0.05, **p < 0.01, and ***p < 0.001. Only significant values are shown (n = 6 animals/group, n = 6 patients). All data supporting figures are provided as source data files.

The online version of this article includes the following source data for figure 2:

**Source data 1.** The numerical data for the graphs in *Figure 2*.

## TDCA/valine treatment induces robust weight loss and ameliorates obesity-related insulin resistance

Based on our metabolomic profiling data, indicating restored systemic TDCA/valine levels after SGx, we next set out to assess the physiological impact of TDCA/valine on obesity and weight loss and administered both metabolites intraperitoneally to naive DIO mice for a course of 2 weeks. Notably, the combined injection of TDCA and valine increased absolute, systemic TDCA and valine concentrations in DIO mice comparable to levels measured in lean control mice (*Figure 3A*), resulting in robust weight loss (*Figure 3B*) that went beyond the observed effects in mice undergoing SGx

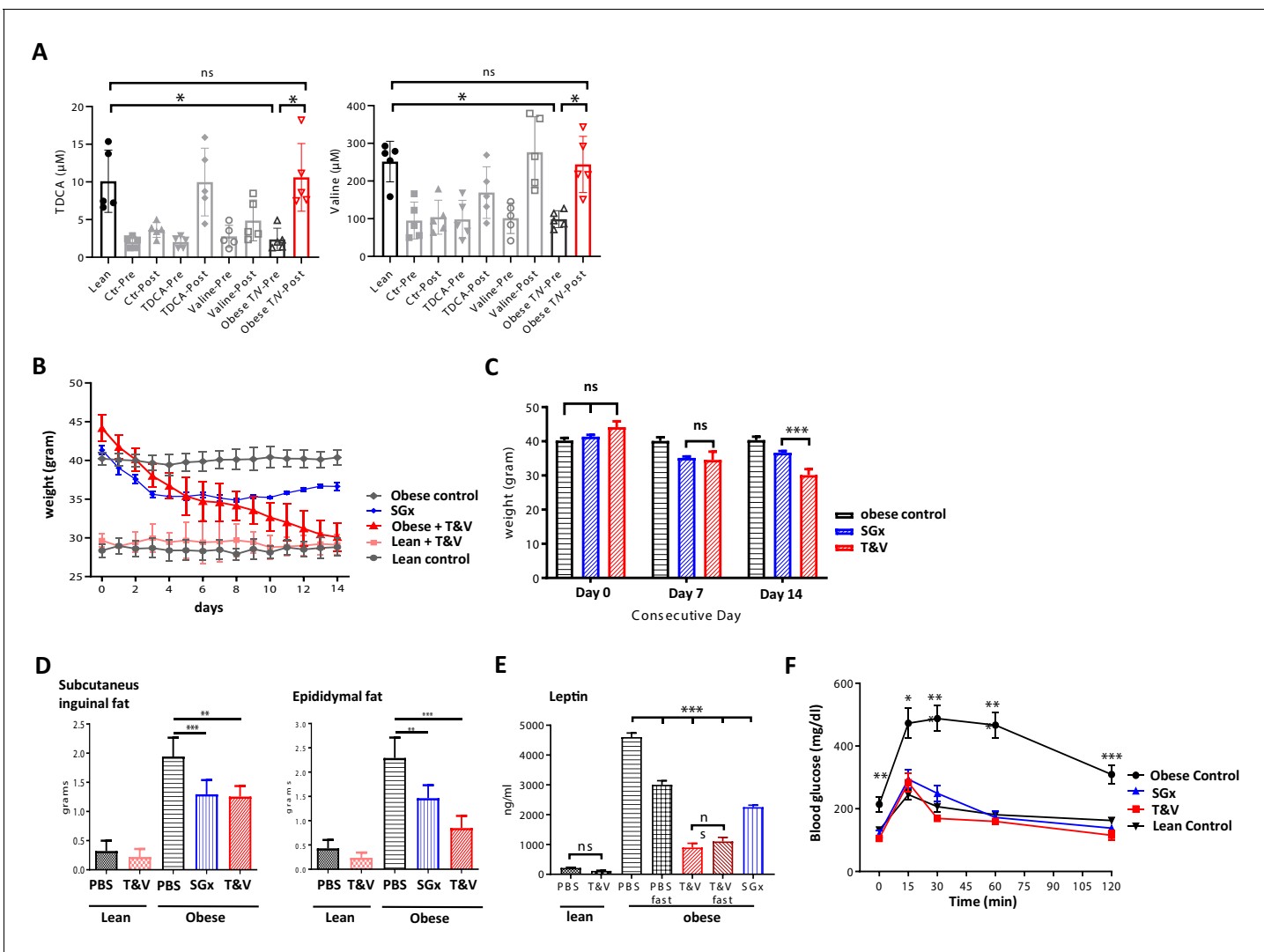

**Figure 3.** Taurodeoxycholic acid (TDCA)/valine treatment induces robust weight loss and ameliorates obesity-related insulin resistance. C57BL/6 diet-induced obese (DIO) mice received intraperitoneal injections of TDCA (50 mg/kg) and L-valine (200 mg/kg) daily over the course of 2 weeks. (A) Plasma TDCA and valine levels were quantified before and after 14 days of treatment by mass spectrometry (n = 5). (B) Body weight was evaluated for 2 weeks every 24 hr. (B) Column plot of mean body weight comparing sleeve gastrectomies (SGx) and TDCA/valine (T/V)-treated animals at days 0, 7, and 14. (C) Subcutaneous and epididymal fat tissue was removed after 14 days of treatment or SGx and weight was determined. (D) Systemic leptin levels were quantified by ELISA after fasting in control and T/V-treated DIO and lean mice. (E) 2 g/kg glucose was injected following 8 hr of daytime fasting. Blood glucose levels were assessed in blood samples utilizing a blood glucose meter. Results are representative of at least three independent experiments. Column plots display mean with standard deviation. Statistical significance was determined by using one-way analysis of variance (ANOVA). Asterisks indicate p-values: *p < 0.05, **p < 0.01, and ***p < 0.001. Only significant values are shown (n = 5–7 animals/group).

The online version of this article includes the following source data and figure supplement(s) for figure 3:

**Source data 1.** The numerical data for the graphs in *Figure 3*.

**Figure supplement 1.** Effect of taurodeoxycholic acid (TDCA), valine, and the combinatorial treatment with TDCA and valine are shown.

(*Figure 3C*). Importantly, administration of TDCA/valine to lean control mice did not impact weight loss (*Figure 3B*). Of note, isolated administration of either TDCA or valine was found to be less effective as assessed by body weight measurements over a course of 30 days with daily, i.p. TDCA or valine injections (*Figure 3—figure supplement 1*).

To further characterize the impact of TDCA/valine, we assessed subcutaneous and epididymal adipose tissue stores and found a significant decrease of both fat stores in TDCA/valine-treated DIO mice comparable to effects observed in DIO mice that underwent SGx (*Figure 3D*).

In addition, we assessed whether treatment with TDCA/valine affected systemic levels of leptin as a main neuroendocrine peptide released by adipocytes (*Brennan and Mantzoros, 2006*). Indeed, our results indicated reduced systemic leptin levels in both, SGx and TDCA/valine-treated animals (*Figure 3E*).

It is well established that obesity promotes insulin resistance and T2D (*Saltiel and Kahn, 2001*; *Xu et al., 2003*). Thus, to further explore the effects of TDCA/valine treatment on obesity-associated insulin resistance and T2D, we assessed the capacity for glucose tolerance. DIO mice treated with TDCA/valine displayed a complete reversal of obesity-related insulin resistance comparable to lean controls. Notably, beneficial metabolic effects subsequent to TDCA/valine were analogous to those observed in DIO mice following SGx (*Figure 3F*).

## Treatment with TDCA/valine induces weight loss through altered feeding behavior in the absence of physical dysfunction

Growing evidence suggests a decreased food intake as critical for the long-term weight reduction subsequent to SGx (*Stefater et al., 2012*). Hypothesizing that TDCA/valine treatment may mimic SGx-induced weight loss, we next assessed calorimetric and metabolic parameters of DIO mice receiving daily injections of either TDCA/valine or PBS using a Comprehensive Lab Animal Monitoring System (CLAMS). While cumulative energy intake and the hourly food intake significantly declined in TDCA/valine-treated animals compared to PBS-treated controls (*Figure 4A,B*), hourly measured energy expenditure and the measured total activity did not change during treatment (*Figure 4B,C*).

Moreover, the respiratory exchange ratio and energy balance of TDCA/valine-treated obese animals significantly declined, supporting the significance of fatty acids for energy metabolism (*Figure 4E,F*). Of note, read-outs had not been impacted by the circadian cycle as comparable results were obtained during light and dark periods. Intriguingly, our data on weight loss induced by TDCA/valine were neither linked to a typical food-seeking behavior in response to fasting nor to unspecific toxic side effects of the applied metabolites. Instead, treatment with TDCA/valine specifically targeted feeding behavior while leaving locomotor activity and energy expenditure unaffected thus generating a negative energy balance.

## TDCA/valine-induced weight loss is mediated through suppression of hypothalamic levels of orexigenic MCH

After detecting reduced feeding behavior and preserved energy expenditure linked to TDCA/valine-associated weight loss, we next analyzed hypothalamic peptides that are involved in regulation of appetite and energy homeostasis while treating animals with TDCA/valine. To this end, we isolated hypothalamus tissue from acute brain slices and analyzed mRNA levels of neuroendocrine regulators (AgRP, CART, MCH, NPY, POMC) by real-time PCR (RT-PCR), comparing expression in fed versus fasted animals. Our analysis revealed a substantial attenuation in the increase of MCH with fasting following TDCA/valine treatment (*Figure 5A*). Obese animals treated with PBS and fasted for 12 hr prior to procurement of hypothalamus tissue showed a pronounced increase in MCH levels, findings that were in line with the orexigenic effects of this neuropeptide. In TDCA/valine-treated obese animals, however, MCH-increase was not observed, suggesting an effect of TDCA/valine on the regulation of hypothalamic MCH expression (*Figure 5A*). Notably, MCH has been shown to centrally promote food intake, augment anabolic energy regulation (*Hervieu, 2003*; *Pissios and Maratos-Flier, 2003*; *Shimada et al., 1998*), and increase body weight in experimental models (*Della-Zuana et al., 2002*).

To analyze the effect of TDCA/valine on MCH-regulated appetite and energy homeostasis in vivo, we made use of an MCHR1-I (*Igawa et al., 2016*). Treatment with MCHR1-I in DIO-obese

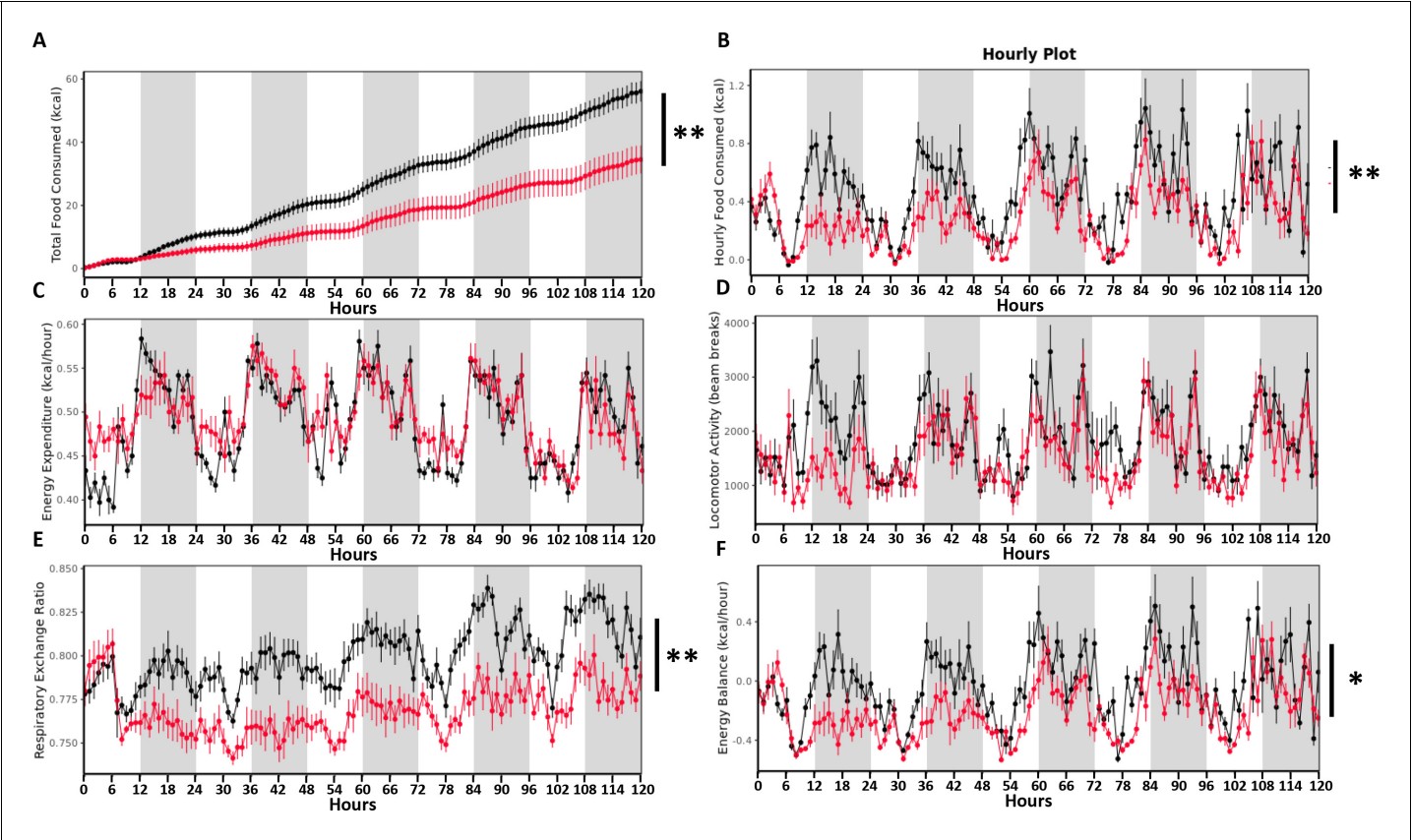

**Figure 4.** Taurodeoxycholic acid (TDCA)/valine treatment induces weight loss through altered feeding behavior in the absence of reduced energy expenditure. Twelve DIO mice (control = 6, treatment = 6) were placed into the Columbus Instruments' Comprehensive Lab Animal Monitoring System (CLAMS) for 6 days. Time graphs represent hourly averages throughout the experiment. Shaded regions represent the 12 hr dark photoperiod. After 1 day of acclimation (not shown), injections of TDCA/valine were performed at 2 p.m. for 5 days. This experiment monitored (A) cumulative energy intake (B) hourly food intake, (C) energy expenditure, (D) locomotor activity, (E) respiratory exchange ratio, and (F) energy balance (energy intake minus energy expenditure). Results are representative of at least three independent experiments. Statistical significance was determined by analysis of variance (ANOVA) using total mass as the covariate. Error bars represent SEM. Asterisks indicate p-values: *p < 0.05, **p < 0.01. Only significant values are shown (n = 6 animals/group).

The online version of this article includes the following source data for figure 4:

**Source data 1.** The numerical data for the graphs in *Figure 4*.

mice led to decreased food intake and subsequent weight loss (*Figure 5B*). More importantly, simultaneous treatment with both MCHR1-I and TDCA/valine did not further reduce food intake or promote additional weight loss in DIO mice, suggesting that effects of TDCA/valine are, at least in part, mediated through suppression of hypothalamic levels of MCH.

Central administration of MCH has been reported to promote food intake in rats (*Della-Zuana et al., 2002*) while peripherally administered MCH has been shown to be unable to pass the blood-brain barrier (*Kastin et al., 2000*). We thus used DIO Wistar rats that underwent cranial surgery, mounting a canula into the lateral cerebral ventricle, thus providing access for intracranial injections. To confirm a specific TDCA/valine-induced inhibition of MCH, we used a combinatorial treatment regimen of i.p.administered TDCA/valine and centrally administered recombinant MCH. Consistent with our findings in mice, obese rats rapidly lost weight under TDCA/valine treatment within 2 weeks (*Figure 5C*). Strikingly, this effect was significantly diminished when co-administering recombinant MCH and TDCA/valine (*Figure 5C*), suggesting an inhibitory effect of TDCA/valine on MCH regulation (*Figure 6*).

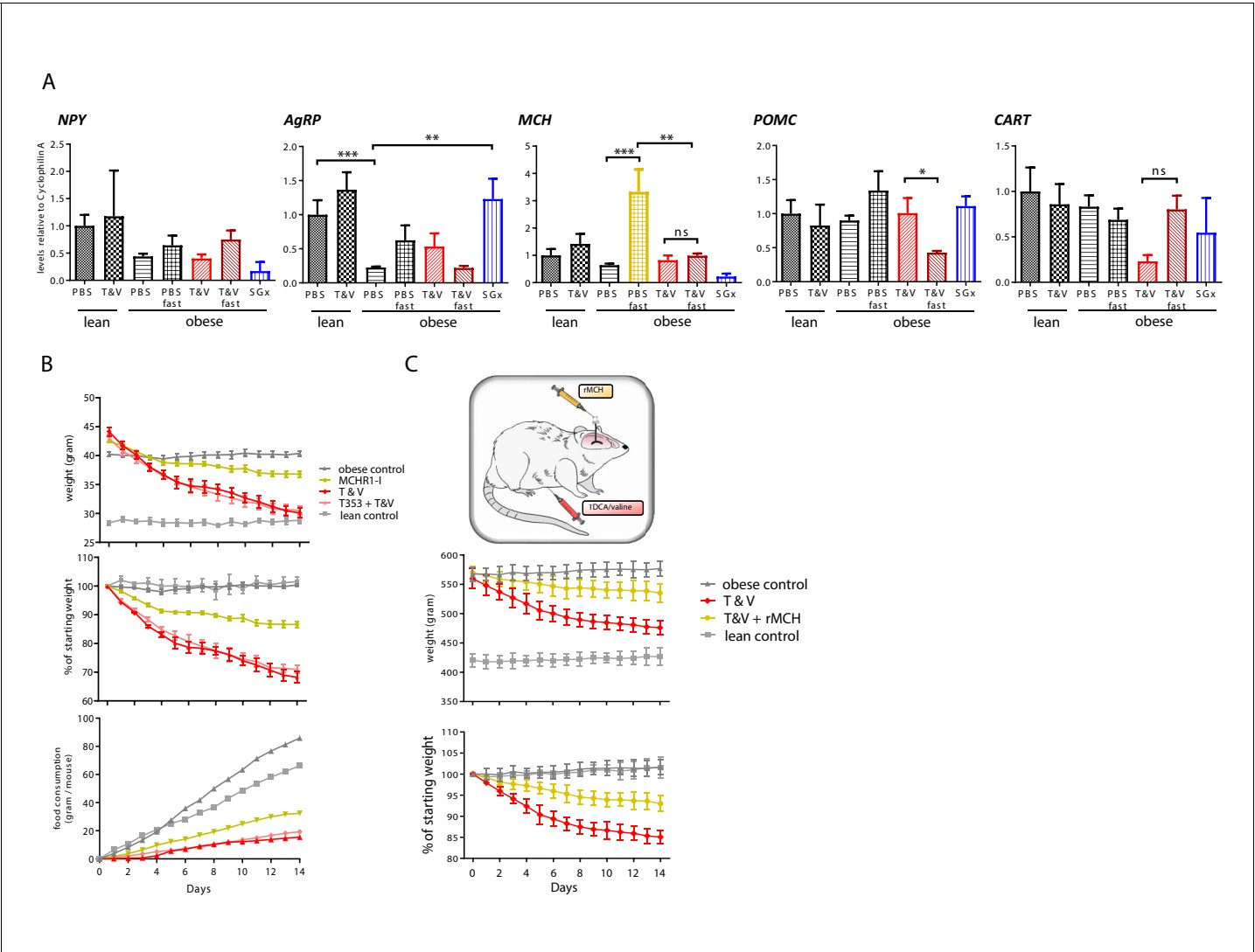

**Figure 5.** Taurodeoxycholic acid (TDCA)/valine treatment acts through suppression of hypothalamic melanin-concentrating hormone (MCH) levels. (**A**) Lean and diet-induced obese (DIO) mice were treated daily with either PBS or T and V. Two groups of T and V-treated DIO mice were subjected to 12 hr fasting before tissue procurement. After 2 weeks, all mice were sacrificed, hypothalamus tissue dissected, and RNA levels of *POMC*, *CART*, *NPY*, *AgRP*, and *MCH* measured by real-time PCR (RT-PCR). (**B**) DIO mice were subjected to daily intraperitoneal (i.p.) injection of T and V, oral administration of MCH receptor one inhibitor (MCHR1-I), or a combined treatment of both, T and V + MCHR1-I for a course of 2 weeks. Body weight and food consumption was measured and expressed as a contingency plot displaying total weight, percentage of starting weight, and food consumption per mouse. (**C**) DIO rats were subjected to combined i.p. TDCA/valine injection and intracerebral administration of recombinant MCH and weight loss was monitored for 2 weeks. Results are representative of at least three independent experiments. Column plots display mean with standard deviation. Statistical significance was determined using one-way analysis of variance (ANOVA). Asterisks indicate p-values: *p < 0.05, **p < 0.01, and ***p < 0.001. Only significant values are shown (n = 5–7 animals/group).

The online version of this article includes the following source data for figure 5:

**Source data 1.** The numerical data for the graphs in *Figure 5*.

## Discussion

Obesity has been associated with major metabolic changes that contribute to obesity-related disorders (*Xu et al., 2003*). In morbid obesity, bariatric surgery is often the only alternative to achieve sustained weight loss. More importantly, among various bariatric procedures, SGx has been shown to induce favorable metabolic changes (*Ryan et al., 2014*; *Patti et al., 2009*; *Pournaras et al., 2012*) while improving insulin sensitivity and glycemic control. Although SGx has made significant advances in reversing obesity, many risks are associated with this surgical

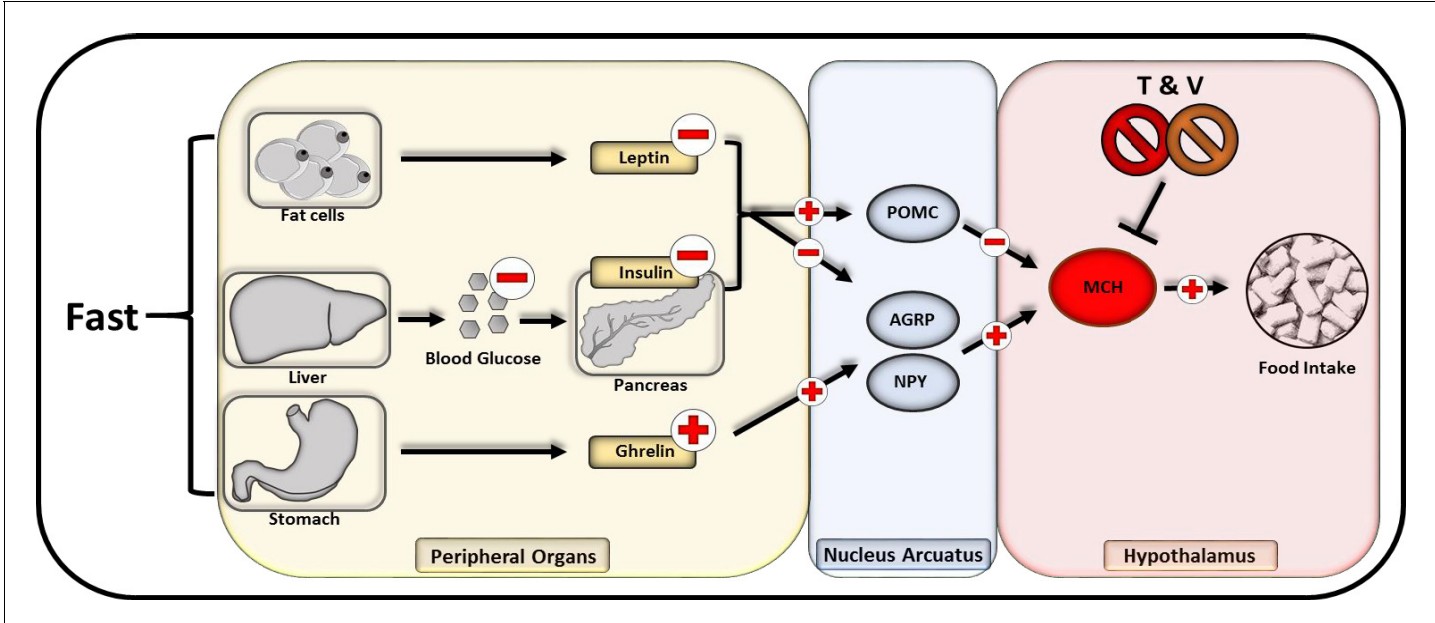

**Figure 6.** Flowchart of neuropeptide-mediated appetite regulation and taurodeoxycholic acid (TDCA)/valine interaction. Fasting promotes the release of melanin-concentrating hormone (MCH) communicated through various endocrinological pathways. During fasting, fat cells decrease the secretion of leptin while a compromised hepatic gluconeogenesis dampens the pancreatic insulin secretion. Within the stomach in turn, an augmented amount of ghrelin is released. All three pathways lead to an activation of AGRP and NPY releasing neurons within the nucleus arcuatus while inhibitory pro-opiomelanocortin (POMC) neurons are impeded. Subsequently AGRP and NPY neurons promote the activation of MCH releasing neurons located in the hypothalamus that are directly stimulating appetite and food intake.

procedure including excessive bleeding, infection, blood clots, and leaks resulting from surgery, thus urging for alternative, non-invasive approaches.

Recently, bile acid signaling has been identified as a mechanistic underpinning of weight loss subsequent to SGx (*Ryan et al., 2014*). Indeed, durable weight loss and improved glucose tolerance are mediated through bile acid-induced FXR signaling, affecting overall food intake and feeding behavior (*Ryan et al., 2014*). Thus, weight loss subsequent to SGx may only in part be communicated through the restricted gastric capacity. Moreover, re-feeding mice that underwent SGx with an HFD revealed that these animals gained weight without reaching pre-surgery obesity, supporting a relevant role of appetite behavior in addition to a modified metabolism (*Stefater et al., 2010*). In contrast, diet-restricted DIO mice restored lost body weight by exhibiting hyperphagia, fully re-gaining their body weight with the end of the diet (*Brownlow et al., 1993*). Hence, a decreased food intake is likely to constitute a primary impetus for the long-term weight loss observed after SGx (*Stefater et al., 2012*). In support, pair-feeding studies have shown that rats restricted to the same daily caloric intake as SGx animals exhibited comparable weight loss (*Stefater et al., 2012*).

These findings are in line with our study, suggesting that durable weight loss and amelioration of obesity-associated disorders after SGx are mediated through both, molecular and central appetite-regulating effects of TDCA/valine. Indeed, our results indicated that TDCA/valine-treated animals had a markedly reduced food intake without altered energy expenditure or locomotor activity, thus excluding toxic side effects of TDCA/valine as a potential confounder of appetite reduction. Notably, TDCA/valine administered to lean animals did not lead to weight loss, emphasizing on appetite-regulating effects only in obese animals while excluding inflammatory processes as confounding side effects. Therefore, our observation that TDCA/valine treatment mimicked metabolic changes that result from SGx are supported by our finding that decreased food intake is a cardinal feature of SGx-induced weight loss.

Within the last decades increased levels of branched-chain amino acids (BCAA) levels have been associated with T2D and obesity-derived complications (*Lynch and Adams, 2014*). Notably, various studies have suggested increased BCAA levels to be a consequence of obesity rather than its cause. Indeed, mitochondrial impairment and obesity-derived low-grade inflammation has been associated

with decreased levels of enzymes that degrade BCAA (*She et al., 2007*; *Burrill et al., 2015*). Moreover, a recent study employing a combination of thiazolidinediones and metformin failed to demonstrate decreasing systemic BCAA levels despite an increased insulin sensitivity (*Irving et al., 2015*). We thus speculate that the weight reducing effect of L-valine in combination with TDCA is not impacting obesity directly but rather exerts effects through a reduction of appetite.

Supporting our results, BCAA have shown beneficial effects on weight reduction. Supplementation of leucine to drinking water of DIO mice reduced body weight by 32% over 10 weeks (*Zhang et al., 2007*). Furthermore, leucine administration has been shown to induce a partial resistance to diet-induced obesity that resulted into reduced hyperglycemia and hypercholesterolemia in DIO mice (*Zhang et al., 2007*). The activation of pro-opiomelanocortin (POMC) neurons in the nucleus arcuatus of the mediobasal hypothalamus (*Blouet et al., 2009*) alters the feeding behavior of DIO mice and has been proposed as a critical mechanistic effect. Activation of mammalian target of rapamycin (mTOR), a signaling pathway integrating sensing of the energy status and growth and proliferation, for instance has been identified as a mechanistic underpinning of BCAA-derived weight loss. Notably, intracerebroventricular administration of leucine decreased food intake leading to body weight loss in DIO rats through mTOR activation of POMC neurons in the nucleus arcuatus while rapamycin, a potent mTOR inhibitor abolished those effects (*Cota et al., 2006*). In the current study, we could not detect significant differences in the expression of POMC in obese and TDCA/valine-treated animals that had fasted prior to hypothalamic procurement. However, POMC neurons exhibit inhibitory effects on neurons of the lateral hypothalamic area in which the orexigenic peptide MCH is localized (*Mystkowski et al., 2000*).

It is well established that MCH regulates feeding behavior, energy balance, and promotes food intake (*Della-Zuana et al., 2002*). In line with those observations, we detected a dramatically augmented MCH expression in obese animals upon fasting that has been absent in both fasted TDCA/valine-treated animals and animals that had been kept on HFD and subjected to TDCA/valine administration. Since MCH is an essential mediator of appetite regulation (*Nahon, 2006*), we hypothesized a potent effect of TDCA/valine on MCH suppression. Consistently, the combinatorial administration of a selective MCHR1 antagonist (*Kawata et al., 2017*) and TDCA/valine failed to elicit additional weight loss, suggesting that TDCA/valine acts at least in part through inhibiting MCH. This hypothesis was confirmed in DIO Wistar rats showing a diminished responsiveness to TDCA/valine treatment when treated intrathecally with recombinant MCH. Of note, TDCA/valine-derived weight loss had not been fully reversed subsequent to intrathecal MCH administration. Thus, model-dependent restrictions in MCH application or dosing aspects may be of relevance. In support, other studies have reported a decreased intensity of MCH-promoted food intake over the course of 24 hr with a peak efficiency 2 hr after administration (*Della-Zuana et al., 2002*; *Rossi et al., 1997*). Moreover, by restoring the bile acid pool (*Ryan et al., 2014*) through TDCA administration, FXR signaling is likely to constitute an additional pathway mediating metabolically derived weight loss. While our studies show that the weight reducing effects of TDCA/valine are communicated, at least in part, through MCH, additional pharmacological and mechanistic studies of TDCA/valine-mediated impact on hypothalamic neuropeptide expression will be necessary. Moreover, future studies will need to determine whether TDCA/valine administration may exert beneficial effects through elevated serum or cerebrospinal fluid levels and if additional metabolic effects altering mitochondrial metabolism and potentially affecting other BCAA levels are in effect.

In summary, our results add critical novel data on mechanisms and consequences of surgery-induced weight loss. More importantly, and of translational relevance, we introduce TDCA and L-valine as novel agents for weight loss and appetite regulation.

## Acknowledgements

The authors wish to express their appreciation and gratitude to Dr Maratos-Flier, Division of Endocrinology, Diabetes and Metabolism, Beth Israel Deaconess Medical Center, Boston, MA, USA, and Novartis Institute for BioMedical Research, Cambridge, MA, USA, for expert advice and helpful discussions. Funding: This work has been supported in part by a grant from NIH (UO-1 A1 132898 to SGT, DP, and MA). MQ was supported by the IFB Integrated Research and Treatment Centre Adiposity Diseases (Leipzig, Germany) and the German Research Foundation (QU 420/1–1). JI was supported by the Biomedical Education Program (BMEP) of the German Academic Exchange Service

(DAAD). TH (HE 7457/1–1) and FK (KR 4362/1–1) were supported by the German Research Foundation (DFG). HRCB was supported by the Swiss Society of Cardiac Surgery. YN was supported by the Chinese Scholarship Council (201606370196) and Central South University. HU, TM, and RM were supported by the Osaka Medical Foundation. CSF was supported by the German Research Foundation (DFG, SFB738, B3)

## Additional information

### Funding

| Funder | Grant reference number | Author |
| --- | --- | --- |
| National Institutes of Health | UO-1 A1 132898 | David Perkins<br>Maria-Luisa Alegre<br>Stefan G Tullius |
| Deutsche Forschungsgemeinschaft | QU 420/1-1 | Markus Quante |
| Deutsche Forschungsgemeinschaft | HE 7457/1-1 | Timm Heinbokel |
| Deutsche Forschungsgemeinschaft | KR 4362/1-1 | Felix Krenzien |
| Chinese Scholarship Council | 201606370196 | Yeqi Nian |
| German Research Foundation | DFG SFB738 B3 | Christine S Falk |
| Osaka Medical Research Foundation for Intractable Diseases | | Tomohisa Matsunaga<br>Hirofumi Uehara<br>Ryoichi Maenosono |
| European Society of Cardiology | | Hector Rodriguez Cetina Biefer |
| German Academic Exchange Service | | Jasper Iske |
| Boston Claude D. Pepper Older Americans Independence Center | 5P30AG031679-10 | Hao Zhou |

The funders had no role in study design, data collection and interpretation, or the decision to submit the work for publication.

### Author contributions

Markus Quante, Timm Heinbokel, Resources, Data curation, Software, Formal analysis, Validation, Investigation, Visualization, Methodology, Writing - original draft; Jasper Iske, Resources, Data curation, Software, Formal analysis, Validation, Investigation, Visualization, Methodology, Writing - original draft, Writing - review and editing; Bhavna N Desai, Formal analysis, Validation, Investigation, Writing - review and editing; Hector Rodriguez Cetina Biefer, Yeqi Nian, Felix Krenzien, Tomohisa Matsunaga, Hirofumi Uehara, Ryoichi Maenosono, Investigation, Writing - review and editing; Haruhito Azuma, Johann Pratschke, Christine S Falk, Tammy Lo, Reza Abdi, Writing - review and editing; Eric Sheu, Data curation, Writing - review and editing; Ali Tavakkoli, David Perkins, Conceptualization, Resources, Data curation, Funding acquisition, Validation, Writing - review and editing; Maria-Luisa Alegre, Conceptualization, Data curation, Funding acquisition, Validation, Methodology, Writing - review and editing; Alexander S Banks, Conceptualization, Data curation, Formal analysis, Supervision, Validation, Investigation, Methodology, Writing - original draft, Writing - review and editing; Hao Zhou, Stefan G Tullius, Conceptualization, Resources, Data curation, Software, Formal analysis, Supervision, Funding acquisition, Validation, Investigation, Methodology, Writing - original draft, Project administration, Writing - review and editing; Abdallah Elkhal, Conceptualization, Data curation, Formal analysis, Supervision, Validation, Investigation, Methodology, Writing - original draft

## Author ORCIDs
Alexander S Banks (ID) http://orcid.org/0000-0003-1787-6925
Hao Zhou (ID) https://orcid.org/0000-0001-9109-2489
Stefan G Tullius (ID) https://orcid.org/0000-0003-3058-3166

## Ethics
Human subjects: Serum samples from patients prior to and 3 months post sleeve gastrectomy were obtained with approval of the Brigham and Women's Hospital (BWH) Institutional Review Board and through cooperation with Dr. Eric G. Sheu and the Center for Metabolic and Bariatric Surgery at BWH. Informed consent was obtained from all patients and samples were collected following BWH ethical regulations.

Animal experimentation: Animal use and care were in accordance with institutional and National Institutes of Health guidelines. The study protocol was approved by the Brigham and Women's Hospital Institutional Animal Care and use Committee (IACUC) animal protocol (animal protocol 2016N000371).

## Decision letter and Author response
Decision letter https://doi.org/10.7554/eLife.62928.sa1
Author response https://doi.org/10.7554/eLife.62928.sa2

# Additional files

## Supplementary files
• Transparent reporting form

## Data availability
All relevant data supporting the findings of this study are available as source data files.

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
