## [Decision Letter]

**Acceptance summary:**

The authors used metabolomics to identify Valine and TDCA as metabolites depleted in diet-induced obesity (DIO) and replenished after sleeve gastrectomies (SGx) in mice. Intraperioneal injection of these two metabolites mimics many of the benefits of SGx, including weight loss, reduced adipose stores and insulin sensitivity. These benefits are related to Val/TDCA's ability to reduce food intake without altering locomotor activity, leading to a negative energy balance. Val/TDCA injection eliminated the fasting-associated rise in hypothalamic MCH expression in obese mice, and central injections of recombinant MCH blunted weight loss induced by Val/TDCA. Overall, this paper suggests a role for Val and/or TDCA in regulating food intake through MCH.

**Decision letter after peer review:**

Thank you for submitting your article "Restored TDCA and Valine Levels Imitate the Effects of Bariatric Surgery" for consideration by *eLife*. Your article has been reviewed by 2 peer reviewers, including Ralph J DeBerardinis as the Reviewing Editor and Reviewer #1, and the evaluation has been overseen by David James as the Senior Editor.

The reviewers have discussed the reviews with one another and the Reviewing Editor has drafted this decision to help you prepare a revised submission.

Summary:

In the paper, the authors used metabolomics to identify Valine and TDCA as metabolites depleted in diet-induced obesity (DIO) and replenished after sleeve gastrectomies (SGx) in mice. Intraperioneal injection of these two metabolites mimics many of the benefits of SGx, including weight loss, reduced adipose stores and insulin sensitivity. These benefits are related to Val/TDCA's ability to reduce food intake without altering locomotor activity, leading to a negative energy balance. Val/TDCA injection eliminated the fasting-associated rise in hypothalamic MCH expression in obese mice, and central injections of recombinant MCH blunted weight loss induced by Val/TDCA. Overall, this paper reports interesting and surprising observations related to the impact of metabolomic disturbances in obesity, and suggests a role for Val and/or TDCA in regulating food intake through MCH.

Essential revisions:

1. It is unclear from the data whether the effects are derived from valine, TDCA, or both. Both reviewers felt that any reader would want to see experiments where either of these metabolites is injected alone.

2. No quantitative metabolite concentration values are provided anywhere, making it difficult to evaluate the robustness of the data. How much do the levels of TDCA and valine change with SGx in mice and humans, and what levels are achieved with the injections of these metabolites in the mice?

---

## [Author Response]

Essential revisions:1. It is unclear from the data whether the effects are derived from valine, TDCA, or both. Both reviewers felt that any reader would want to see experiments where either of these metabolites is injected alone.

We appreciate this comment and have now included novel in vivo experiments in which we treated high fat diet (HFD)-induced obese mice with TDCA alone, valine alone or both TDCA+valine (Supp. Figure 1).

2. No quantitative metabolite concentration values are provided anywhere, making it difficult to evaluate the robustness of the data. How much do the levels of TDCA and valine change with SGx in mice and humans, and what levels are achieved with the injections of these metabolites in the mice?

We agree with the reviewers that absolute concentrations of TDCA and valine constitute essential data. Therefore, we have now quantified absolute concentrations of TDCA and valine in lean, DIO and DIO mice that were treated with TDCA and/or valine. The data have been now included in Figure 3A.